# The Effects of Sn Doping MnNiFeO_4_ NTC Ceramic: Preparation, Microstructure and Electrical Properties

**DOI:** 10.3390/ma15124274

**Published:** 2022-06-16

**Authors:** Dongcai Li, Cangbao He, Ranran Wu, Haiyan Xu, Fengjun Zhang

**Affiliations:** 1Anhui Key Laboratory of Advanced Building Materials, Anhui Jianzhu University, Hefei 230022, China; hecangbao@std.ahjzu.edu.cn (C.H.); wrr2000@stu.ahjzu.edu.cn (R.W.); 2Key Laboratory of Functional Molecule Design and Interface Process, Anhui Jianzhu University, Hefei 230601, China; fjzhang@ahjzu.edu.cn

**Keywords:** NTC ceramic, low temperature solid state reaction, microstructure, electrical properties, Sn-doping

## Abstract

Sn-doped MnNiFeO_4_ ceramic with negative temperature coefficient (NTC) was prepared through the low-temperature solid-phase reaction route (LTSPR), aiming at improving the sintering behavior and modulating the electrical properties. The experimental results of the ceramic powder precursor indicate that the calcination of the ceramic precursors at above ~300 °C is an exothermic process, which contributes to the transition of the ceramic powder from the amorphous phase into the crystal spinel phase; the spinel phase of ceramic powders can be formed initially at ~450 °C and well-formed at ~750 °C. A high densification of ~98% relative densities and evenly distributed grains within an average size of 2~12 μm for the sintered Sn-doped specimen were obtained. The specific resistance and B-value were notably increased from 12.63 KΩ·cm to ~24.65 KΩ·cm, and from 3438 K to ~3779 K, respectively, with the Sn-doping amount. In contrast, the aging rates of the Sn-doped specimen have not changed markedly larger, waving around ~2.7%. The as-designed Sn-doped MnNiFeO_4_ can be presented as a candidate for some defined NTC requirements.

## 1. Introduction

Negative temperature coefficient (NTC) ceramic thermistors based on transition metal oxides, are widely applied in temperature measuring, controllers, time-delay devices, infrared detectors, and voltage regulators [1,2,3,4,5,6]. NiMn_2_O_4_ based spinel-type NTC ceramic thermistors have been researched widely for their modifiable structure and properties [7,8,9,10,11,12,13,14,15,16,17,18,19,20,21,22,23]. For pure NiMn_2_O_4_, the homogeneous spinel phase is obtained with difficultly due to the NiO phase segregation occurring above 950 °C during sintering in air, and separation of α-Mn_2_O_3_ and the ilmenite type NiMnO_3_ during cooling below 730 °C. In contrast, ternary or plural transition metals systems can form more stabilized spinel structures, especially for ternary Mn-Ni-Fe-O_4_ or Mn-Co-Fe-O_4_ systems [5,7]. The element doping strategy for modifying the structure and properties are often adopted. Tetravalent ions Zr^4+^, Si^4+^, Sn^4+^ and Ti^4+^ doped in Mn_3−x−y_Ni_x_Fe_y_O_4_ were studied, in order to modify properties of the spinel thermistors by B-site ion substitution in the spinel structure, and the specific resistivity and sensitivity were increased [19,20,21,22,23]. Doping strategies of trivalent ions In^3+^, Al^3+^ and Cr^3+^ [13,14,15,16,17,18] and bivalent ions Cu^2+^, Zn^2+^ and Mg^2+^ [9,10,11,12] were reported as well, which all modulated the thermistor properties by the B-site ion substitution. It is well-accepted that the conductivity mechanism for transition metal oxide thermistors belongs to a thermally activated phonon-assisted hopping mechanism of charge carriers between Mn^4+^ and Mn^3+^ on crystallographically equivalent lattice B sites in the spinel structure. As a consequence, the mobility of the charge carriers is relatively low and is described as small polarons conduction [1]. 

In this study, MnNiFeO_4_ ceramic composition was optioned as the host matrix owing to its relatively stable properties and low cost, and Sn^4+^ was the selected modifier due to its bigger ionic radius with a larger ion polarity, aiming at improving the sintering behavior and modifying the electrical properties. Two Sn-doping strategies of Mn_1−x_Sn_x_NiFeO_4_ and isomolar Sn-doping substitution for Mn, Ni, and Fe in MnNiFeO_4_, respectively, were designed. The samples were prepared by the low-temperature solid-phase reaction route. The phase formation of the ceramic powders, microstructure and electrical properties of the sintered samples were studied in detail.

## 2. Materials and Methods

NTC ceramic Mn_1−x_Sn_x_NiFeO_4_ (x = 0, 0.03, 0.06, 0.09), MnNi_0.91_Sn_0.09_FeO_4_ and MnNiFe_0.91_Sn_0.09_O_4_ (designated as S1~S6 specimen, see Table 1) were prepared by the low-temperature solid-phase reaction route. The experimental process chart is shown in Figure 1. The analytical grade starting materials of H_2_C_2_O_4_·2H_2_O, Mn(CH_3_COO)_2_·4H_2_O, Ni(CH_3_COO)_2_·4H_2_O, FeCl_2_·4H_2_O, and SnCl_2_·2H_2_O were used for synthesizing ceramic precursors. The compounds for introducing ceramic constituent cations and oxalic acid were weighed accurately according to the molar ratio 3:3 of the total metallic cations to H_2_C_2_O_4_·2H_2_O. The weighed batches were mixed and ground in agate mortar by hand for 50~70 min until yellow green paste ceramic powder precursors were obtained. The obtained yellow green pastes were dried at 75 °C for 30 h, and then ground again. The as-prepared precursor powders were heated to 750 °C at a heating rate of 5 °C/min and kept for 600 min, and the ceramic powders were acquired with particle sizes in the submicrometer range. A 5% PVA solution was used for granulation of fine calcined powders. The green compacts were formed by pressing the granulated powder in a uniaxial press to a size of 5 mm diameter × 2 mm thick discs. The green compacts were sintered at 1250 °C for 300 min in air and cooled naturally.

The dried precursor powders were characterized by thermogravimetric analysis and differential scanning calorimetry (TG-DSC, Universal V4.5A) and Fourier transform infrared spectroscopy (FTIR, Nicolet 6700,Thermo Fisher, USA). The phase purity of the calcined powders and sintered samples was analyzed using X-ray diffraction (XRD, D8 Advance, Bruker, Germany). The unit cell dimension was calculated from XRD data by least-squares refinement. Scanning electron microscopy (SEM, Gemini SEM450, ZEISS, Germany) was used for the microstructural characterization of the sintered pellets. X-ray photoelectron spectra (XPS, ESCALAB 250, ThermoVG Scientific, USA) analysis was performed to identify the presence of different ions in the crystal sublattice of spinel structure. Monochromatic Al Kα was chosen as the radiation source in the XPS instrument and operated at 11 kV and 10 mA. For electrical characterization, the sintered discs were silvered using a thermistor grade silver paste and annealed at 850 °C for 40 min in a small muffle furnace. The electrical resistance was measured at 25 °C and 85 °C, respectively, using self-made equipment with a temperature uniformity ± 0.1 °C using a 6_1/2_ precise digital multimeter (34401A, Agilent, USA). The tested pellets were subjected for reliability studies by giving an accelerated aging at 150 °C for 1000 h, and the resistance measurements for aged specimens were repeated. There were 5 specimens per composition for electrical test.

## 3. Results

### 3.1. TG-DSC Analysis of the Ceramic Precursors

The TG-DSC curves of S1 and S4 ceramic precursors (green paste dried at 75 °C for 30 h) heated to 800 °C at 10 °C/min in air are shown in Figure 2 and Figure 3, respectively. It can be seen that, at lower temperature stages of free water and adsorbed water removal [24,25], the weight loss of the as-dried S1 precursor is about 5.37% at 20~124.31 °C, whereas that of the S4 precursor is 15.72% + 3.67% = 19.39% at 20 °C~101.04 °C~147.58 °C. The weight loss of S4 is higher than that of S1 by 14.32% at this stage, in other words, S4 precursor has more free water and adsorbed water. This is because the as-prepared precursor paste of S4 has a higher viscosity and dries with more difficulty at 75 °C.

At elevated temperature stages responsible for crystal water removal, the weight loss of S1 precursor is 19.78% at 124.31~242.06 °C, whereas that of S4 precursor is 15.22% at 147.58~235.71 °C. Theoretically, the crystal water content of S1 precursor MnNiFe (C_2_O_4_)_3_⋅4H_2_O is 16.6%; that of S4 precursor Mn_0.91_Sn_0.09_NiFe (C_2_O_4_)_3_⋅4H_2_O is 16.4%. 

At higher temperature weight-loss stages responsible for oxalate decomposition, the weight loss of S1 precursor is 32.91% at 242.06~342.18 °C, and that of S4 precursor is 26.94% at 235.71~364.04 °C. Exclusive of the effect of free water and adsorbed water, the loss molar ratio of the C_2_O_4_^2−^ to crystal water at the last two stages for S1 and S4 is respectively 6.67% and 6.27%, which demonstrates that Sn-doping has a certain but minor effect on precursor preparation. 

The corresponding DSC curves show that the peak at 209.23 °C for S1 precursor and the peak at 207.41 °C for S4 precursor are clearly endothermic peaks due to crystal water removal. Moreover, S4 precursor has a larger endothermic peak below 100 °C, which indicates again that SnCl_2_·4H_2_O introduction into the precursor batches changed the precursor paste viscosity and kept more free water during the drying process. Both S1 and S4 precursors demonstrate large exothermic peaks at 311.36 °C and at 331.23 °C, respectively, which should be ascribed to the air oxidation of the decomposed product of as-synthesized oxalate. It is interesting that there is a little exothermic peak, respectively, for both S1 and S4 exiting at around 500 °C with a weight loss of around 2%, which is considered as the oxidation of carbon black resulting from the oxalate decomposition. The exothermic phenomena at above 300 °C are considered as an advantage for spinel phase forming from the amorphous state of the ceramic precursor due to the internal enhanced temperature. 

### 3.2. FTIR Results of S1 and S4 Precursor Powders

Figure 4 shows the FTIR spectra for the S1 and S4 ceramic precursors dried at 75 °C for 30 h. The broadbands at around 3389.5 cm^−1^ for the two samples are assigned to hydroxyl stretching band of water molecules; 1640 cm^−1^ is assigned to C=O vibration. The minor strong bands that are centered at 1306 cm^−1^ and 1356 cm^−1^ for both S1 and S4 precursors are assigned to the symmetric ν(O-C-O) stretching vibration of the oxalate, and 1443 cm^−1^ only occurring to S4 precursors could be assigned to the asymmetric ν(O−C=O) stretching band induced by Sn-doping. The band appearing at 818 cm^−1^ is assigned to the bending vibration of asymmetric δ(O-C-O) [26]. The band at 476 cm^−1^ should be associated to the stretching vibration of M-O (M: Mn, Ni, Fe, Sn).

### 3.3. X-ray Results of the Calcined Powders and Sintered Samples

The XRD patterns of the samples of the calcined S4 Precursors at 450 °C, 500 °C, 550 °C and 750 °C are shown in Figure 5. It can be seen that the spinel structure of the as-synthesized S4 ceramic powder is initially formed at 450 °C, and well-formed at 750 °C, whereas the single spinel structure phase by the conventional solid-state reaction preparation method is well-formed at least at 1000 °C for Sn-doped manganese-based NTC composition due to the larger Sn ion radius. The structures of all the sintered samples at 1250 °C are single cubic spinel structures shown in Figure 6. The lattice parameters of S1~S4 Mn_1−x_Sn_x_NiFeO_4_ (x = 0, 0.03, 0.06, and 0.09) in Table 2 increase slightly with the increase of Sn-doping amount due to the larger Sn ion radius, and the lattice parameters exhibit nearly no change for the samples S4, S5 and S6 with isomolar Sn-doping substitution for Mn, Ni, and Fe in MnNiFeO_4_. Based on the theoretical densities, the calculated relative densities of all as-prepared specimen are larger than 96% (see Table 2). 

### 3.4. SEM and EDS Element Maps of Sintered Samples

The surface microstructures of the sintered samples S1~S6 are shown in Figure 7. It can be seen that the effect of Sn-doping is very obvious on the surface morphology, which appear quite different among all the samples. The surface of S1 consists of dome-shaped grains with smooth line-shape boundaries, the grain size distribution is between 2~8 μm, and there are some pores measuring 2~4 μm. The surface of S2 consists of uniformly alternate arrangement of bigger plain-plate-shape grains of 6~10 μm and convex smaller facet grains of 2~6 μm. The surface of S3 has a similar grain arrangement as that of S2, but the surface of S3 is flatter and has a groove-shaped grain boundary. More specially, the surface of S4 consists of uniform little convex grains with notably projected grain boundaries, which should be attributed to the decreased sintering activation energy induced by Sn insertion in the lattice, leading to the near-melting-point sintering for Sn aiding the sintering effect, resulting from the larger ion polarization. It is very interesting and worthy of further studying that the projected grain boundary is still in a crystal state (see the insert with 20,000 of S4 in Figure 7). The surface of S5 (MnNi_0.91_Sn_0.09_FeO_4_) is cleanest and smoothest, which consists of uniform plain-plate-shape grains of 2~12 μm with a concave grain boundary. In contrast, S6 (MnNiFe_0.91_Sn_0.09_O_4_) has more large grains of 8~14 μm, and the grain boundary also exhibits a concave shape.

From the morphologies of S1 to S4 (Mn_1−x_Sn_x_NiFeO_4_ (x = 0, 0.03, 0.06, 0.09), associated with the densification degree of sintered samples (see Table 2), it can be summarized that the effect of minor Sn-doping favors sintering densification due to the effect of Sn-doping on lowering sintering temperature. Comparing the morphologies of S4 to S6 (Sn isomolar substitution for Mn, Ni and Fe in MnNiFeO_4_), it can be concluded that the effect of Sn-substituted for Ni in MnNiFeO_4_ is the best, which coincides with the aging electrical property. 

In order to further interpret the morphology of the S4 sample, the element maps were constructed by EDS scanning (Figure 8). It was not found that the segregation of Sn ions occurred on the grain boundary as expected initially, and Sn ions dissipated uniformly in the whole body. The other elements also distribute uniformly, which demonstrates that all the elements demonstrate good solubility within the as-designed composition ranges.

### 3.5. XPS Results of Sintered S1 and S4 Samples

The occupation of the cations in the spinel structure represents an interesting subject for many researchers, since the physical properties (magnetic, electrical, catalytic, etc.) of these materials strongly depend on the distribution of the cations in the lattice [27]. Various methods have been used to study the cation distribution, such as XRD, neutron diffraction, infrared (IR) spectroscopy, Mössbauer spectroscopy and XPS [28,29,30,31,32], of which XPS is the most valid and convenient method. In Ni_0.6_Mn_2.4−x_Sn_x_O_4_ spinels, the Mn^3+^, Mn^4+^ and Ni^2+^ ions prefer to occupy the B-site, while Mn^2+^ ions tend to reside at the A-site. The location of the Sn^4+^ ions preferred the B-site to the A-site in the spinel structure which was analyzed by XRD [23]. In this study, XPS was adopted to show Sn valence and location in MnNiFeO_4_. The binding energy of Sn3d_5/2_ in S4 is 486.5 eV, which is consistent with 486.6 eV of SnO_2_ [32], and is identified by the deconvolution fitting of Sn3d_5/2_ in Figure 9d. The identified Sn ions with +4 valence state in MnNiFeO_4_ prefer to locate to oxygen octahedral sites according to the valence principle, which will affect the amount of Mn^3+^/Mn^4+^ charge pairs. It can be seen from Figure 9b,c that Mn^3+^:Mn^4+^ ratio of 1.218 for S1 is slightly larger than that of 1.189 for S4. This indicates that Sn located in B-sites leads to Mn^3+^/Mn^4+^ charge pairs dropping down and resulting into the increase of specific resistivity for S4 (see Figure 10).

### 3.6. Electrical Properties of the As-Prepared NTC Ceramics

The specific resistivity and B-values of S1~S6 tend to increase with the increase of Sn-doping amount, in which S5 has a maximum specific resistivity of 32.62 KΩ·cm and a maximum B-value of 3928 K. The specific resistivity for S1~S4 Mn_1−x_Sn_x_NiFeO_4_ (x = 0, 0.03, 0.06, and 0.09) increases monotonously with the amount of Sn substituting for Mn. This can be explained by the B-occupied Mn^3+^/Mn^4+^ pairs being decreased by the introduced Sn^4+^ ions entering into B-sites, which coincides with the results of XPS. In contrast with S4 (MnNi_0.91_Sn_0.09_FeO_4_) and S6 (MnNiFe_0.91_Sn_0.09_O_4_), S5 (MnNi_0.91_Sn_0.09_FeO_4_) has the largest resistivity, which can be explained by the Sn substitution for Ni in nominal composition decreases the content of Ni^2+^ locating B-sites, which can yield Mn^4+^ in the B-site from valence balance. In contrast, S4 (Sn nominal substitution for Fe) and S6 (Sn nominal substitution for Mn) exhibit no more obvious effects on specific resistivity. All the B-values of S1~S6 show the same trend with the specific resistivity, which complies with the common rules for the spinel NTC thermistors.

The aging behavior of NTC ceramics in the system Mn-Ni-Fe-O are dependent on composition and the oxidation which occurs during the cooling after sintering [33]. In the spinel structure, described by AB_2_O_4_, the distribution of the ions over the oxygen interstitial sites: Mn^3+^ will occupy predominantly the B-site (oxygen octahedral interstitial) while Mn^2+^ will be on the A-site (oxygen tetrahedral interstitial) and almost all Ni^2+^ will go to the B-site [31]. At high temperatures, during sintering, there is a tendency for disordering due to the entropy principle. Upon cooling, reordering of the cations of the A- and B-sites occurs. During aging a cationic vacancy migration occurs from grain boundary to bulk. A cation-vacancy-assisted migration of cations occurs to thermodynamically more stable sites [33]. It was reported that the stoichiometrically cooled samples in N_2_ atmosphere showed a much smaller aging rate than the air cooled samples [34].

The average aging rates ΔR_25°C_/R_25°C_ of all as-prepared samples no longer show a difference and waver around 2.7%, in which S2 and S3 have slightly less average aging rates (see Figure 11 and Table 3). According to the Shannon ionic table [35], the solid ionic radius of Sn^4+^ under hexa-/tetra-coordination number situation is 0.83 Å/069 Å, and that of Mn^4+^ under hexa-coordination situation is 0.67 Å; that of Mn^3+^ under hexa-coordination situation is 0.72 Å and 0.758 Å at low-spin and high-spin state, respectively; that of Ni^2+^ under hexa-/tetra-coordination is 0.83 Å/0.69 Å; that of Fe^3+^ under hexa-/tetra-coordination is 0.78 Å/0.63 Å.

It is simply the larger radius of Sn^4+^ that induces the pinning effect under oxygen hexa-coordination inhibiting the exchange of the A-site and B-site ions; on the other hand, the higher densification contributes to oxygen penetration during the aging at the elevated temperature of 150 °C, leading to the stoichiometry uniformity in the whole grain volume. The higher densification also favors stoichiometry uniformity of the grains during cooling. S4~S6 demonstrate larger aging rates of ~3.08% for more Sn introduction. The largest aging rate of S4 is ascribed to excess Sn introduction of 9% molar resulting in the larger lattice distortion, yielding lattice stress. In contrast with S4 and S6, S5 has a relatively small aging rate, which should be owed to the good sintering behavior, leading to a higher relative density with a uniform microstructure.

## 4. Conclusions

Sn-doped MnNiFeO_4_-based NTC ceramics with a single spinel structure were prepared successfully by LTSPR. The TG-DSC and XRD results of as-synthesized ceramic powders indicate that, at above ~300 °C, the calcination of the ceramic precursors is an exothermic process, favoring a solid-state reaction; the spinel phase of ceramic powder is formed initially at ~450 °C and well-formed at ~750 °C. A high densification of ~98% relative densities and evenly distributed grains with an average size of 2~12 μm for the sintered doping Sn specimen can be obtained simply by the conventional process. The improved sintering behavior and distinct microstructures for the Sn-doped specimen can be attributed to the sintering promoted effect of the larger radius Sn ion inclusion. The specific resistance and B-value with Sn-doping amount are notably increased from 12.63 KΩ·cm to ~24.65 KΩ·cm, and from 3438 K to ~3779 K, respectively, which can be attributed to the decrease of Mn^4+^/Mn^3+^ pairs due to Sn^4+^ substitution for Mn^4+^ in the spinel B sites. In contrast, the aging rates of the Sn-doped specimens have not changed markedly, waving around ~2.7%, which can be ascribed to the combined effect of Sn ion inclusion: advantageous effects lie in the pinning effect from inhibiting the exchange of the A-site and B-site ions, and the oxidation inhibition effect during the aging and cooling process for the higher densification with a more uniform microstructure, and the disadvantageous effect lies in the induced lattice stress by the introduction of larger radius Sn ions. The Sn-modified MnNiFeO_4_ with improved sinterability presents a promising candidate for some defined preparation and property requirements.

## Figures and Tables

**Figure 1 materials-15-04274-f001:**
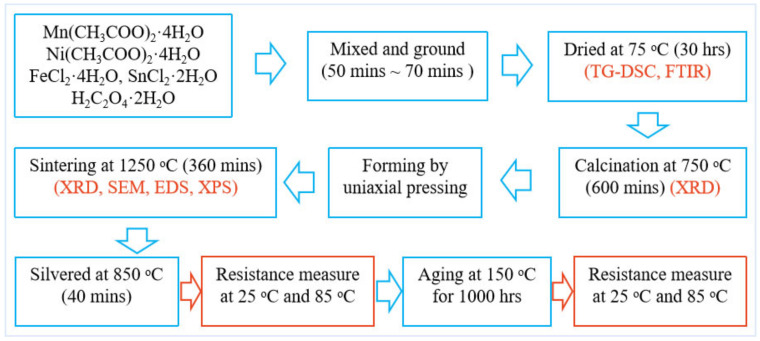
Illustration of the experimental process of the ceramic specimen.

**Figure 2 materials-15-04274-f002:**
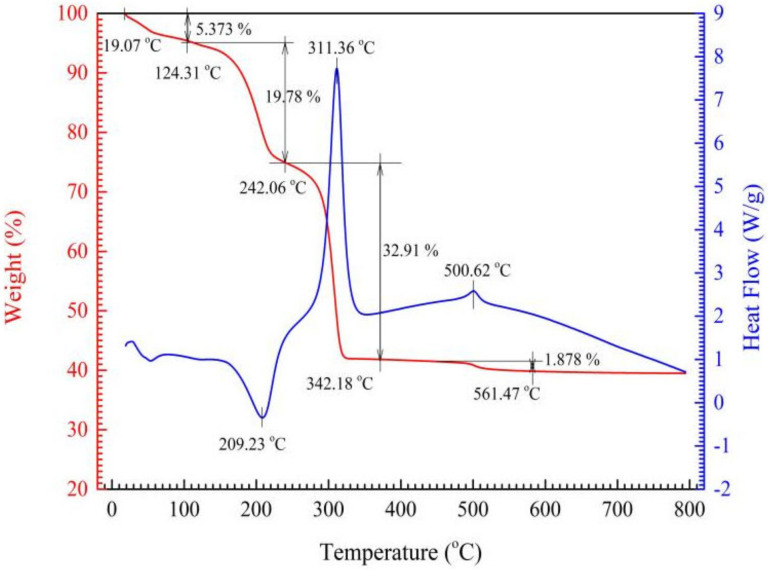
TG and DSC curves of S1 ceramic precursor.

**Figure 3 materials-15-04274-f003:**
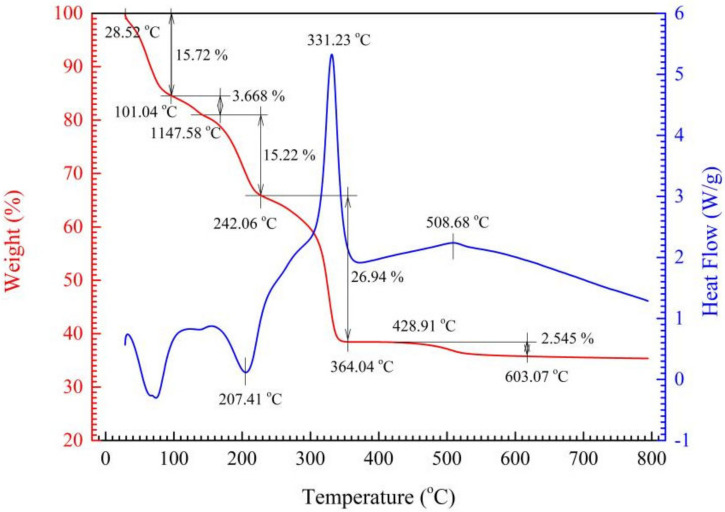
TG and DSC curves of S4 ceramic precursor.

**Figure 4 materials-15-04274-f004:**
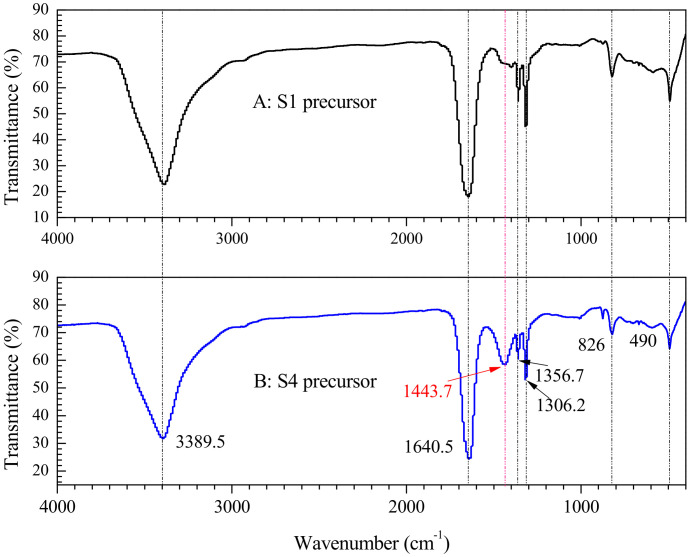
FTIR of dried S1 ceramic precursor (**A**) and dried S4 ceramic precursor (**B**).

**Figure 5 materials-15-04274-f005:**
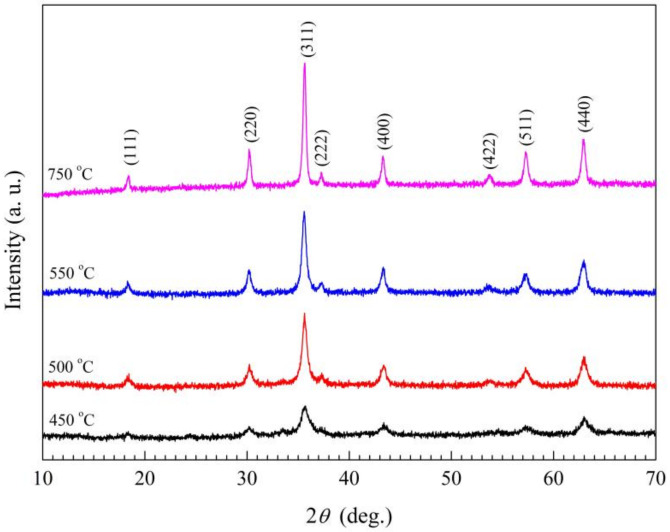
The XRD patterns of the calcined powders of S4 ceramic precursors.

**Figure 6 materials-15-04274-f006:**
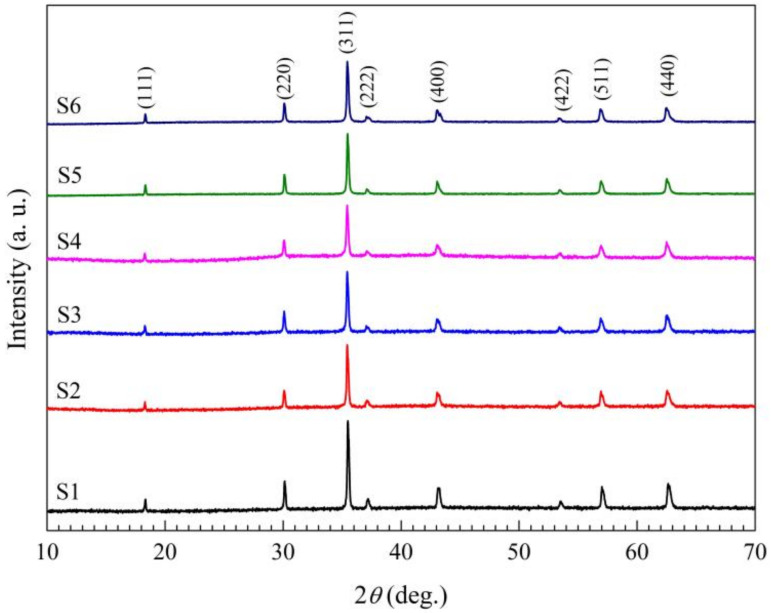
The XRD patterns of S1~S6 samples sintered at 1250 °C.

**Figure 7 materials-15-04274-f007:**
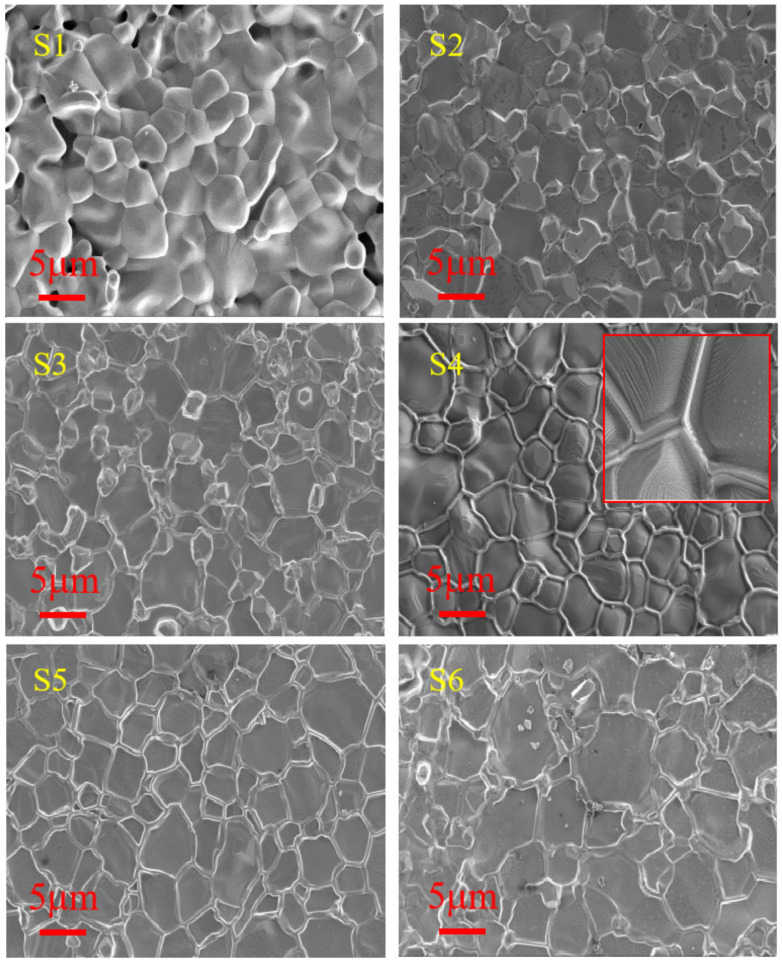
SEM images of the sintered S1~S6 sample surfaces.

**Figure 8 materials-15-04274-f008:**
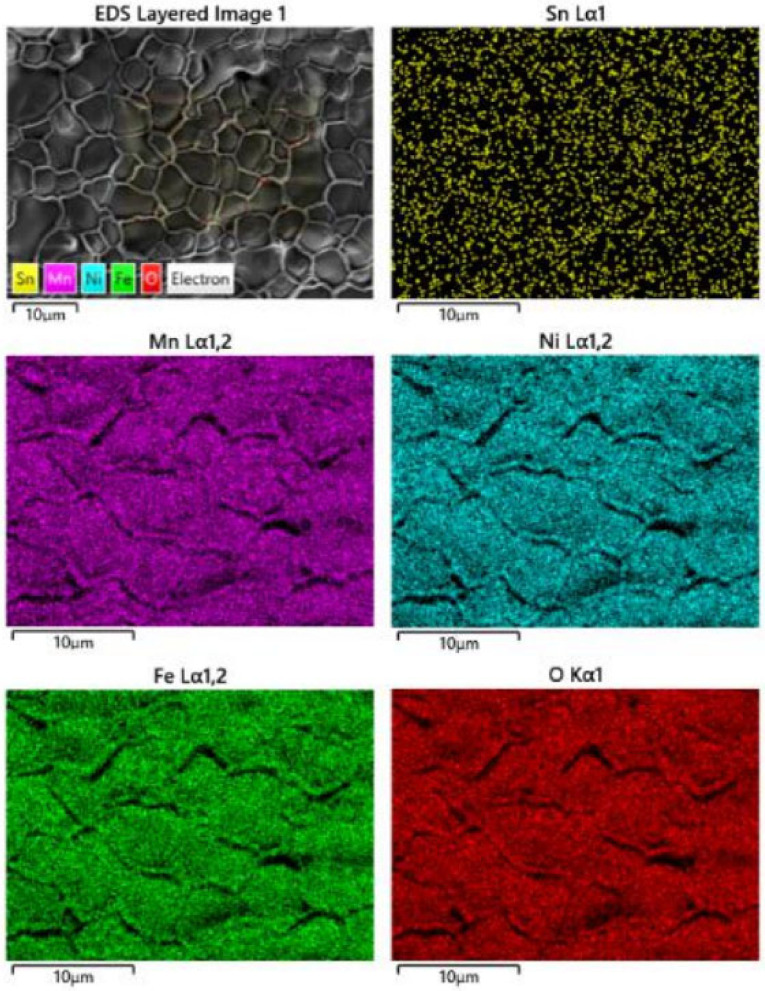
The EDS element maps of the sintered S4 sample surfaces.

**Figure 9 materials-15-04274-f009:**
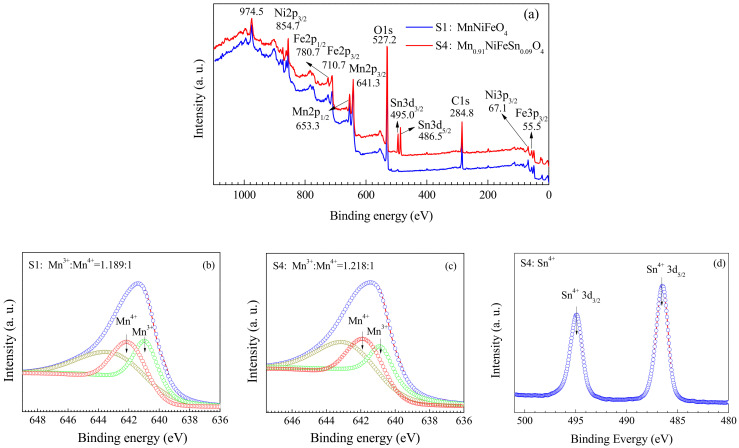
(**a**) XPS survey spectra of the sintered S1 and S4; (**b**,**c**) XPS fitting curves of Mn2P for S1 and S4, respectively; (**d**) XPS fitting curve of Sn3d for S4.

**Figure 10 materials-15-04274-f010:**
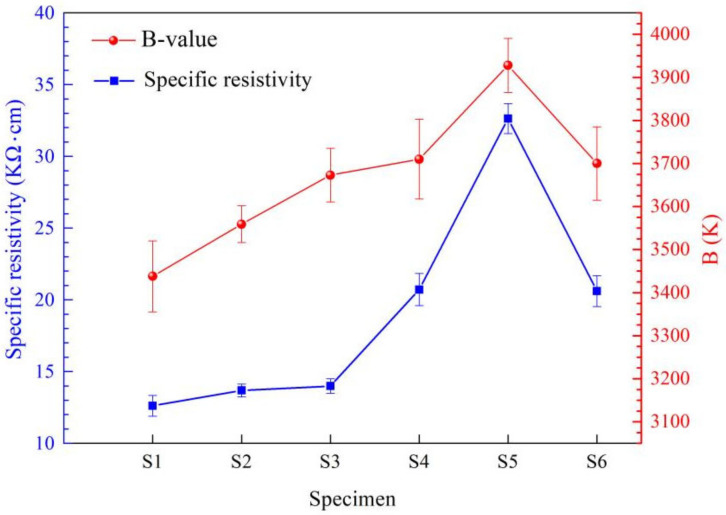
The resistivity and B-values of the sintered samples of S1~S6.

**Figure 11 materials-15-04274-f011:**
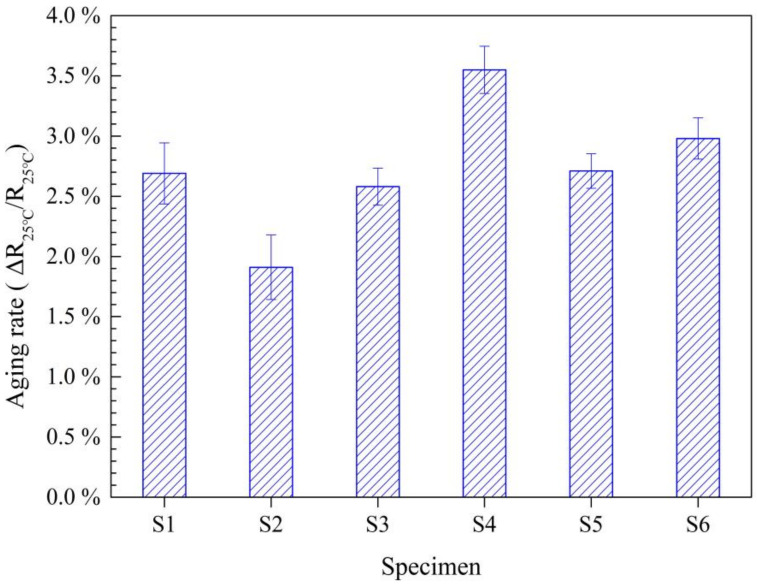
The aging rates of the sintered samples of S1~S6.

**Table 1 materials-15-04274-t001:** The composition of Sn-doped MnNiFeO_4_ ceramics.

Compositions	Mn_1−x_Sn_x_NiFeO_4_				MnNi_0.91_Sn_0.09_FeO_4_	MnNiFe_0.91_Sn_0.09_O_4_
	X = 0.0	X = 0.03	X = 0.06	X = 0.09		
Designated	S1	S2	S3	S4	S5	S6

**Table 2 materials-15-04274-t002:** The lattice parameters and densities of the sintered samples.

Specimen	S1	S2	S3	S4	S5	S6
Lattice parameters (Å)	8.3828	8.3831	8.3832	8.3834	8.3833	8.3834
Theoretical densities (g·cm^−3^)	5.265	5.308	5.351	5.394	5.386	5.392
Relative densities (%)	96.2	98.9	97.8	97.1	98.8	96.9

**Table 3 materials-15-04274-t003:** The resistivity, B-values and aging rates of the sintered samples.

Samples	S1	S2	S3	S4	S5	S6	Mn_1.46_Ni_0.54_FeO_4_ Ref. [33] ^1^
Resistivity (KΩ·cm)	12.63	13.69	13.99	20.72	32.62	20.61	~22
B-value (K)	3438	3559	3673	3710	3928	3700	~4600
Aging rate ΔR_25°C_/R_25°C_ (%)	2.69	1.91	2.58	3.55	2.71	2.98	~2.6

Note ^1^: The data are estimated by measuring the curves of Figure 1 and Figure 2 in Ref. [33].

## Data Availability

Not applicable.

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
