# Peer review of "The Effects of Sn Doping MnNiFeO4 NTC Ceramic: Preparation, Microstructure and Electrical Properties"

_materials, 2022, doi:10.3390/ma15124274_

Round 1
Reviewer 1 Report
In this manuscript, The Effects of Sn Doping MnNiFeO4 NTC Ceramic such as preparation, microstructure and electrical properties were investigated. The manuscript is of great importance in the related field of research. But, the way in which the results were presented should be improved much. Some questions and suggestions are provided below to enrich the scientific level of the presented manuscript before its publication:
- The abbreviations such as NTC for the first time should be introduced.
- The quality of the figures, especially for the ones contain the numerical results, can also be enhanced.
- The introduction must be enriched by adding relevant references.
- The conclusion was be improved.
Author Response
dear reviewer:
Now i have finished the revision. the anwers as following:
1.The abbreviations such as NTC for the first time should be introduced.
The abbreviations of NTC,TG-DSC, FTIR, XRD, SEM, EDS and XPS have been introduced at the first time appearance.
2. The quality of the figures, especially for the ones contain the numerical results, can also be enhanced.
The figures 1,2 containing numbers have been redrawn, and other figures are also touched.
3.The introduction must be enriched by adding relevant references.
The introduction adds five and more relevant references,especially in the year 2009 afterwards.
4.The conclusion was be improved.
The conclusion has been rewritten
Thanks.
Reviewer 2 Report
In this study, MnNiFeO4 ceramic composition with relatively stable properties was optioned as the host matrix. Two Sn doping strategies of Mn1-xSnxNiFeO4 and iso-molar Sn-doping substitution for Mn, Ni, and Fe in MnNiFeO4 were designed. While the topic is interesting, the paper's novelty is not apparent. The experiment procedure and apparatus specification should be explained in more detail, and the result and discussion are not easy to understand. The paper language is below the standard of a reputable international journal.
Therefore, I recommend the publication needs a major revision and provide the following comments are considered in a revised version.
1. I suggest that the abstract need be rewritten. Start the abstract with the significance of the research, followed by methods, conclusions, and future recommendations.
2. I would strongly advise the author to rewrite their introduction by explaining the problem, identifying the previous research gap, and clearly presenting the objective and novelty.
3. The equipment specification and accuracy also the experiment procedure should be explained in detail. Moreover, it is better to present the flow chart for the experimental procedure.
4. A discussion section should describe the research's major findings and compare them to earlier studies. In addition, the current research area's gaps must be highlighted, and future study directions must be presented.
5. The conclusions are not written in a technical manner.
6. I suggest engaging in professional proofreading to restructure the paper language.
Author Response
Dear reviewer:
Now , i have finished my revision. my reply as following:
- I suggest that the abstract need be rewritten. Start the abstract with the significance of the research, followed by methods, conclusions, and future recommendations.
The abstract has been rewritten according to the revised opinion.
- I would strongly advise the author to rewrite their introduction by explaining the problem, identifying the previous research gap, and clearly presenting the objective and
The introduction has been rewritten according to the advice.
- The equipment specification and accuracy also the experiment procedure should be explained in detail. Moreover, it is better to present the flow chart for the experimental procedure.
The flow chart for the experimental procedure has been added. The problem of equipment specification and accuracy is related mainly to a set of self made equipment for the electrical resistance measurement,which has been indicated in the artical with a temperature uniformity ±0.1 oC using a 61/2 precise digital multimeter (Agilent 34401A, USA).
- A discussion section should describe the research's major findings and compare them to earlier studies. In addition, the current research area's gaps must be highlighted, and future study directions must be presented.
The most related earlier studies of the subject is literature 23, with which the gap lies in the difference of composition and preparation. The other related researches of the effect of composition-modified on electrical properties has been cited and discussed. The authors think that further research of relationship of microstructure and electrical properties for NTC ceramic is necessary.
- The conclusions are not written in a technical manner.
The conclusions have been rewritten.
- I suggest engaging in professional proofreading to restructure the paper language.
The paper language has been improved by a experienced professor
thanks
Reviewer 3 Report
Dear Authors,
Include some of the minor improvements for enriching the quality of the manuscript.
Figures 4 and 5 should clearly incorporate crystal structure in the XRD peak and brodly explain each crystal structure.
Figures 6 and 7 should mention all the components in the SEM image for immediate attraction to the readers and explain the mechanism with suitable literature support.
Binding energy support is required for at least any of the important properties in Figure 8.
Figures 9 and 10 should incorporate an error bar chart.
Author Response
dear reviewer:
Now, i have finished my revision, my reply as following:
1.Include some of the minor improvements for enriching the quality of the manuscript.
Figures 4 and 5 should clearly incorporate crystal structure in the XRD peak and broadly explain each crystal structure.
2.The as-synthesized powders and sintered samples are single phase with cubic spinel structure, the appeared peaks in the shown XRD patterns all should be assigned to the single phase.
Figures 6 and 7 should mention all the components in the SEM image for immediate attraction to the readers and explain the mechanism with suitable literature support.
Yes, the fascinating SEM images motivate the authors to write the paper. However, the suitable literatures have not been found and get a good verification. The authors just gave a soundable explanation.
3.Binding energy support is required for at least any of the important properties in Figure 8.
Binding energy analysis identified that Sn ions with +4 valence state in MnNiFeO4 prefer to locate oxygen octahedral sites from valence principle, which will affect the amount of Mn3+/Mn4+ charge pairs. It can be seen from Fig. 8(b) and Fig. 8(c) that Mn3+:Mn4+ ratio of 1.218 for S1 is slightly larger than that of 1.189 for S4. It indicates that Sn located in B-sites leads to Mn3+/Mn4+ charge pairs dropping down and resulting into the increase of specific resistivity for S4,which is shown in Fig.9.
4.Figures 9 and 10 should incorporate an error bar chart.
The error bar in Figures 9 and 10 has been added.
thanks
Reviewer 4 Report
Dear Authors, thank you for your manuscript, submitted to "Materials". After the reading the manuscript, I have the following questions and comments:
1. The manuscript contains very old references: there is one reference only, related to 2016-2022 period. The absence of the actual references in the text gives the impression of the irrelevance of the work itself. It is wishable to add the modern references, related to 2019-2022 years.
2. The use of the abbreviation NTC should be clarified once, probably.
3. From the Introduction Part it is not clear, why is the necessity to modify the existing NTC thermistors? Why was Sn chosen as a doping element?
4. Ref. [6] is not relevant in context: it does not contain the M4+ dopants.
5. It is not clear the using of Ref. [17] - it does not contain the data on the band at 1443 cm-1.
6. Why do Figures 4,5 have the non conventional captions of Intensity Axis?
7. "The phase structures.. in Fig. 4". It is the terminological error. For example, It would be written as: The XRD patterns of the samples were shown in Fig.4. The samples were single-phase with cubic spinel structure. And so on..
8. Table 1 does not contain lattice parameters, as mentioned in the text.
9. It is wishable to add the ionic radii of the another cations, mentioned in discussion, besides Sn. The value of 0.71 A for Sn in the text - what is the valence of the cation? Its coordination number? What is the correlation between the radius of Sn and the temperature of 1000 oC, mentioned in the text?
10. Table 2 contains wrong caption (it is related really to the absent Table 1, probably)
11. The resulting Part requires the discussion and comparison with existing literature data, for example, it is wishable to add the resistivities and aging rates values for another NTC thermistors to the Table 2.
Author Response
dear reviewer:
Now i have finished my revision, my reply as following:
1.The manuscript contains very old references: there is one reference only, related to 2016-2022 period. The absence of the actual references in the text gives the impression of the irrelevance of the work itself. It is wishable to add the modern references, related to 2019-2022 years.
The references have been refreshed, and added some latest related references.
- The use of the abbreviation NTC should be clarified once, probably.
The abbreviation NTC has been clarified at the first appearance.
- From the Introduction Part it is not clear, why is the necessity to modify the existing NTC thermistors? Why was Sn chosen as a doping element?
the necessity to modify the existing NTC MnNiFeO4 thermistors lies in improving the sinterability and modifying the electrical properties. Initially, Sn was choosed as a doping element into the MnNiFeO4 aiming at improving the sinterability via the lattice doping or forming liquid sintering. Finally, we found that the doped samples still have a relatively good aging properties, a rational explanation was proposed.
- [6] is not relevant in context: it does not contain the M4+dopants.
Ref. [6] has been rearranged.
- It is not clear the using of Ref. [17] - it does not contain the data on the band at 1443 cm-1.
Ref. [17] has been modulated.
- Why do Figures 4,5 have the non conventional captions of Intensity Axis?
the conventional captions (A.U.) of Intensity Axis have been substituted for the captions of Intensity Axis in Figures 4,5.
- "The phase structures.. in Fig. 4". It is the terminological error. For example, It would be written as: The XRD patterns of the samples were shown in Fig.4. The samples were single-phase with cubic spinel structure. And so on..
"The phase structures.. in Fig. 4" has been revised into “The XRD patterns of the samples of the calcined S4 Precursors at 450 oC, 500 oC, 550 oCand 750 oC were shown in Fig.4. The samples were single-phase with cubic spinel structure.”
- Table 1 does not contain lattice parameters, as mentioned in the text.
Table 1 has added the lattice parameters, as mentioned in the text.
- It is wishable to add the ionic radii of the another cations, mentioned in discussion, besides Sn. The value of 0.71 A for Sn in the text - what is the valence of the cation? Its coordination number? What is the correlation between the radius of Sn and the temperature of 1000 oC, mentioned in the text?
The valence and coordination number for 0.71 Å for Sn in the text has been changed into 0.83Å from shannon ionic table, and the oxygen haxa-coordination number situation have been indicated.the ionic radii of Mn have been added. Mn4+ under oxygen haxa-coordination situation is 0.67Å, Mn3+ under oxygen haxa-coordination situation is 0.72 Å and 0.73 Å,respectively, at low spin and high spin.
- Table 2 contains wrong caption (it is related really to the absent Table 1, probably)
The correct caption for Table 2 has been made, which is “The Resistivity, B-value and Aging rates of the sintered samples”
- The resulting Part requires the discussion and comparison with existing literature data, for example, it is wishable to add the resistivities and aging rates values for another NTC thermistors to the Table 2.
The referred literature data for MnNiFeO4 from Feltz”s Paper in Table 2.
thanks.
Round 2
Reviewer 1 Report
The revised manuscript can be accepted.
Author Response
Dear Reviewer:
Thanks for your acceptance and your hard work for the review.
the author,
Dongcai Li
Reviewer 2 Report
The authors have already accommodated the reviewer's recommendation. There are a lot of improvements in the revised version. Overall, the paper is well written. Therefore, I suggest that the paper could be accepted for publication.
Author Response
Dear reviewer:
Thanks for you acceptance and your hard work for the review.
the author:
Dongcai Li
Reviewer 4 Report
Dear Authors, thank you for your attention to my comments! The manuscript was improved. Now, please, pay your attention on texts' proofreading - some spaces disappeared (for example, see Figure 1). And I would like to propose to expand the notes of B-value and aging rates in Table 3 - they need in temperature range, and from [33] especially.
Author Response
Dear reviewer:
The self-proofreading has been made again,the word spaces have been modulated. One note for Table3 has been added.
thanks,
the author, Dongcai Li